# Insights into the Origin of Activity Enhancement via Tuning Electronic Structure of Cu_2_O towards Electrocatalytic Ammonia Synthesis

**DOI:** 10.3390/molecules29102261

**Published:** 2024-05-11

**Authors:** Meimei Kou, Ying Yuan, Ruili Zhao, Youkui Wang, Jiamin Zhao, Qing Yuan, Jinsheng Zhao

**Affiliations:** School of Chemistry and Chemical Engineering, Liaocheng University, Liaocheng 252059, China; kmm2018705344@163.com (M.K.); yuanyingsdlc@163.com (Y.Y.); zhao1784308@163.com (R.Z.); m19861904240@163.com (Y.W.)

**Keywords:** electronic structure, Cu_2_O, electrocatalytic ammonia synthesis, NO_3_^−^RR, sewage treatment

## Abstract

The insight of the activity phase and reaction mechanism is vital for developing high-performance ammonia synthesis electrocatalysts. In this study, the origin of the electronic-dependent activity for the model Cu_2_O catalyst toward ammonia electrosynthesis with nitrate was probed. The modulation of the electronic state and oxygen vacancy content of Cu_2_O was realized by doping with halogen elements (Cl, Br, I). The electrocatalytic experiments showed that the activity of the ammonia production depends strongly on the electronic states in Cu_2_O. With increased electronic state defects in Cu_2_O, the ammonia synthesis performance increased first and then decreased. The Cu_2_O/Br with electronic defects in the middle showed the highest ammonia yield of 11.4 g h^−1^ g^−1^ at −1.0 V (vs. RHE), indicating that the pattern of change in optimal ammonia activity is consistent with the phenomenon of volcano curves in reaction chemistry. This work highlights a promising route for designing NO_3_^−^RR to NH_3_ catalysts.

## 1. Introduction

Industrially, ammonia is usually prepared by the preparative Haber process, which operates at a temperature of 300–500 °C and a pressure of 3–6 MPa. The process consumes large amounts of non-renewable resources, generates environmental pollution, and releases large amounts of CO_2_ [1,2,3,4]. One of the most harmful consequences is the release of nitrates, which can leach into the soil, contaminate groundwater and surface water, and lead to the eutrophication of water bodies, causing severe disruptions to ecosystems [5,6]. On the other hand, carbon dioxide is a greenhouse gas and poses a significant threat to the environment [7,8]. The method of electrocatalytic ammonia synthesis with NO_3_^−^ has gradually attracted extensive attention and research at home and abroad. The advantages of electrocatalytic ammonia synthesis are as follows: (i) H_2_O is used instead of H_2_ as the hydrogen source, which can reduce the use of fossil fuels; (ii) also, since H_2_O is non-toxic, non-hazardous, and widely available, this reduces environmental risks; (iii) the electricity used in electrocatalytic ammonia synthesis can also come from renewable energy sources such as photovoltaic, wind, and tidal energy [9,10,11,12]. The electrocatalytic nitrate reduction (NO_3_^−^RR) synthesis of ammonia has the potential for application, as shown in Figure 1.

The reduction of nitrate (NO_3_^−^) to ammonia (NH_3_) is a complex reaction containing multiple electron transfers and multistep hydrogenation, accompanied by various by-products. It faces the dual challenges of activity and selectivity. Therefore, owing to the core role of catalysts during the NO_3_^−^RR process, the rational design of effective electrocatalysts with high activity, selectivity, and stability is always the research hotspot [13]. Among various catalysts, copper (Cu)-based catalysts have attracted a lot of research attention due to their relatively high NO_3_^−^RR activity and NH_3_ selectivity, as well as low cost [14,15]. Inspired by the biological heterogeneous reduction of nitrate in nature, and using Cu and Co elements to mimic the adsorption catalytic center and the proton-providing center of the enzyme, respectively, Sun et al. designed Cu-Co alloy nano-sheet catalysts, which have achieved relatively excellent catalytic activity [16]. Zhang et al. investigated the active centers of Cu-based catalysts (Cu_2_O) during NO_3_^−^RR. They found that the Cu^0^/Cu^+^ serves the active phase, and the electron transfer from Cu^+^ to Cu^0^ at the interface of Cu_2_O facilitated the formation of *NOH intermediate and suppressed the hydrogen evolution reaction, leading to high selectivity and faradaic efficiency [17]. Jiang et al. investigated the effect of metal doping Ni, Co, and Fe on copper-based catalysts in electrocatalytic ammonia synthesis. They found that the bimetallic catalysts exhibited higher NH_3_ yields than the monometallic catalysts, which was attributed to the strong synergistic effect between the Cu and other metal components [18]. Wang et al. reported that the Fe single atom catalyst effectively prevents the N-N coupling step required for N_2_ due to the lack of neighboring metal sites, promoting ammonia product selectivity. This catalyst had a maximal ammonia Faradaic efficiency of ~75% and a yield rate of up to ~20,000 μg h^−1^ mg_cat_^−1^ [5]. However, the electroreduction process always induced the morphology and electronic structure evolution of Cu-based catalytic materials, which would cause uncontrollable effects on the electrocatalytic ammonia synthesis reaction. Therefore, unveiling the origin of the Cu active phase during the NO_3_^−^RR process is significant for both the fundamental understanding and rational design of nitrate-to-ammonia electrocatalysts.

Previously, we have uncovered the origin of the facet-dependent activity for the model Cu_2_O catalyst toward urea electrosynthesis, demonstrating that the NO_3_^−^RR activity depends strongly on the crystal facets in Cu_2_O [19]. Further, knowing that Cu_2_O can also serve as a model catalyst due to its adjustable electronic structure, the origin of the prominent activity enhancement via tuning the electronic state of Cu_2_O was investigated. Herein, the modulation of the electronic state and oxygen vacancy content of Cu_2_O was realized by doping with halogen elements (Cl, Br, I). The highest ammonia yields were found at the specific active site of Cu_2_O/Br, with an average ammonia yield of 11.4 g h^−1^ g^−1^ at −1.0 V (vs. RHE), exceeding most of the previous reports. Firstly, the intrinsic activity of its ammonia synthesis was characterized and calculated, and it was found that Cu_2_O/Br has a larger electrochemically active surface area and exposes more active sites. Secondly, through in-depth analysis of the electronic states of Cu_2_O, it was found that as the number of electron-deficient states of Cu_2_O increases, its ammonia-synthesizing performance rises, reaches an optimal state, and then decreases. Finally, an in-depth analysis of the oxygen vacancy content of Cu_2_O halogen doping reveals that the ammonia synthesis performance of Cu_2_O increases with the oxygen vacancy content, with Cu_2_O/Br showing the best reactivity.

## 2. Results and Discussion

### 2.1. Characterization of Catalysts

#### 2.1.1. XRD Analysis

X-ray diffraction (XRD) analysis was conducted to elucidate the crystalline structure of halogen-doped Cu_2_O catalysts, as depicted in Figure 2. The XRD spectra for all specimens exhibited three distinct peaks approximately at 43°, 50°, and 73°, corroborating the successful synthesis of Cu_2_O [20,21]. Upon comparing the peaks of undoped Cu_2_O with those of Cu_2_O/Cl, it was observed that the positions of the three peaks remained unaltered. This phenomenon suggests the uniform dispersion of Cl^−^ into the cuprous oxide matrix. In contrast, the comparison between Cu_2_O and Cu_2_O/Br indicated the presence of three additional characteristic peaks at 2θ ≈ 28°, 44°, and 53° alongside the original peaks at 2θ ≈ 43°, 50°, and 73°, implying successful doping of Br into Cu_2_O [22]. Similarly, the analysis of Cu_2_O versus Cu_2_O/I revealed not only the original three characteristic peaks of Cu_2_O but also four novel peaks at 2θ ≈ 26°, 30°, 42°, and 49°, which signifies the effective incorporation of I into the Cu_2_O lattice [23].

#### 2.1.2. XPS Analysis

The surface valence states of the halogen-doped catalysts were analyzed using XPS analysis. As seen in Figure 3a, it can be seen that Cu_2_O consists of four elements, namely O and Cu. At the same time, the Cu_2_O/X (X = Cl, Br, I) catalysts also contain another three elements, Cl, Br, and I, respectively, indicating the successful doping of halogens into the Cu_2_O. The concentration of each halogen in the Cu_2_O–halogen catalysts was also made by XPS (Appendix A). In Figure 3b, the Cu 2*p* fine spectrum shows two peaks with binding energies of 932.3 eV and 952.6 eV, attributed to Cu^0^/Cu^+^ 2*p*_3/2_ and Cu^0^/Cu^+^ 2*p*_1/2_, respectively, typical XPS binding energy peak positions of Cu^+^, with vibrationally excited companion peaks belonging to Cu^2+^ seen at 942.5 eV and 962.0 eV, indicating the presence of copper oxide on the surface [24,25,26,27]. The presence of CuO in the materials may be due to two reasons: firstly, the samples were stored for a long time before the test, leading to surface oxidation of Cu_2_O to CuO; secondly, residual particulate impurities on the surface of Cu_2_O contained a small amount of Cu^2+^ substances. Comparison of the energy spectra of Cu_2_O and Cu_2_O/X (X = Cl, Br, I) showed a shift in the main peak of Cu, Cu^0^/Cu^+^ 2*p*_3/2_, illustrating that halogen doping affects the electronic state of Cu_2_O and then changes its catalytic activity. The order of influence on the electronic state of cuprous oxide was as follows: Cu_2_O < Cu_2_O/Br < Cu_2_O/I < Cu_2_O/Cl. Additionally, the high-resolution O 1*s* spectrum of pure Cu_2_O (Figure 3c) can be divided into three peaks centered at 530.3, 531.5, and 532.4 eV, attributed to lattice oxygens, oxygen vacancies (OVs), and surface-adsorbed oxygen species (e.g., -OH groups), respectively [28]. The percentages of oxygen vacancies (OVs) in Cu_2_O, Cu_2_O/Cl, Cu_2_O/Br, and Cu_2_O/I were 31.03%, 20.50%, 43.17%, 38.50%, respectively. The content of oxygen vacancies in Cu_2_O showed an increasing and then decreasing trend as the halogen-doped electronegativity decreased, with the Br-doped catalyst having the highest content of oxygen vacancies. Finally, the valence states of halogens in Cu_2_O/X (X = Cl, Br, I) were analyzed. Figure 3d shows the fine spectrum of Cl 2*p* in Cu_2_O/Cl, with Cl 2*p* divided into two main peaks, Cl 2*p*_1/2_ (199.70 eV) and Cl 2*p*_3/2_ (198.1 eV), by Gaussian peak splitting [29,30]. Figure 3e shows the fine spectrum of Br 3*d* in Cu_2_O/Br, with Br 3*d* divided into two main peaks, Br 3*d*_5/2_ (69.1 eV) and Br 3*d*_3/2_ (70.1 eV), by Gaussian peak splitting [31]. Figure 3f shows the fine spectrum of I 3*d* in Cu_2_O/I, with I 3*d* divided into two main peaks, I 3*d*_5/2_ (620.0 eV) and I 3*d*_3/2_ (631.5 eV), by the Gaussian splitting method [32].

#### 2.1.3. SEM Mapping Analysis

The morphology of the catalysts was systematically investigated using Scanning Electron Microscopy (SEM) mapping analysis, as shown in Figure 4. Observations from the SEM images reveal that the incorporation of halogens significantly alters the morphology of the catalysts, leading to an evolution towards more irregular shapes, along with an apparent increase in the fluffiness of the particle size. Figure 4 illustrates a homogeneous distribution of copper (Cu), oxygen (O), and halogen elements, underscoring the uniform doping of halogens into the Cu_2_O crystals. This uniform distribution indicates excellent dispersion and stability across the synthesized catalyst series [33]. High-resolution SEM and TEM images of the synthesized Cu_2_O–halogen catalysts have also been added in Appendix A. Clearly, both Cu_2_O and Cu_2_O/X (X = Cl, Br, I) exhibit irregular shapes, and the doping of halogens has little effect on the Cu_2_O morphology.

### 2.2. Electrocatalytic Performance

Electrochemical investigations were conducted using a standard H-cell reactor configured with a three-electrode system and operated under a fixed potential. The initial phase of the study involved evaluating the potential performance of ammonia electrosynthesis through linear scanning voltammetry (LSV), as depicted in Figure 5a. In solutions containing a mix of nitrate and bicarbonate, the catalysts demonstrated higher current densities compared to those in solely bicarbonate solutions. This suggested the involvement of nitrate ions (NO_3_^−^) in the electrocatalytic reactions. Subsequently, the study focused on the ammonia production capabilities of Cu_2_O and Cu_2_O/X (X = Cl, Br, I) series catalysts. A UV-visible spectrometry sampling method was employed to quantify the ammonia content, as shown in Figure 5b. Specifically, the Cu_2_O/Br catalyst was examined under electrocatalytic conditions using a reversible hydrogen electrode (RHE) across a potential range of −0.4 to −1.4 V in 0.2 V increments. The results, illustrated in Figure 5c, indicated a progressive increase in ammonia yield for the Cu_2_O/Br catalyst. Furthermore, the ammonia selectivity of this catalyst, across a potential range of −0.4 to −1.4 V (vs. RHE), displayed a pattern of initial growth followed by a decline. Additionally, as shown in Figure 5d, the Faradaic efficiency (FE) of ammonia for the Cu_2_O/Br catalyst exceeded 60% within the potential range of −0.8 to −1.2 V, peaking at 62.78% at −1.0 V (vs. RHE), which demonstrated that the catalysts possessed a relatively excellent selectivity. The ammonia yield tended to increase and decrease as the halogen-doped electronegativity decreased, reaching an optimal yield of 11.4 g h^−1^ g^−1^ at Br doping, as depicted in Figure 5e. Figure 5f illustrates that the optimal ammonia FE at −1.0 V was 47.16%, 35.78%, and 43.02% for Cu_2_O, Cu_2_O/Cl, and Cu_2_O/I, respectively [34]. The stability electrocatalysis experiments of the prepared catalysts were carried out and confirmed the excellent electrochemical stability of Cu_2_O/Br (Appendix A). Moreover, a thorough comparison with existing methodologies or materials for electrocatalytic ammonia synthesis was made in this paper. Clearly, the Cu_2_O/Br has excellent ammonia synthesis rates (exceeding most reported catalysts) and appropriate Faraday efficiencies (Appendix A). 

### 2.3. Relationship between Electronic Structure and Activity of Catalysts

To elucidate the intricate relationship between the electronic state of cuprous oxide (Cu_2_O) and its electrocatalytic efficacy in ammonia synthesis, this study embarked on a comprehensive characterization and quantification of the intrinsic activity of Cu_2_O for ammonia production, as delineated in Figure 6. Employing the cyclic voltammetry technique, we meticulously determined the catalysts’ electrochemically active surface area (ECSA), each uniquely doped with varying halogens. This was achieved by extracting the double-layer capacitance (C_dl_) values based on established methodologies [35]. Notably, these C_dl_ values exhibited a direct linear proportionality with the electrochemical surface area of the respective electrodes [36]. Figure 6 represents the capacitive current curves and cyclic voltammograms (CV) for the pristine Cu_2_O and its halogen-doped variants Cu_2_O/X (X = Cl, Br, I). Additionally, the insets within the figure present a detailed view of the capacitive current curve and CV profiles, measured in a potential range of −0.30 to −0.60 V relative to the reversible hydrogen electrode (RHE). These cyclic voltammograms were explicitly recorded in a potential domain where the current response was exclusively ascribed to the charging of the electrical double layer. A pivotal observation from Figure 6 is the discernible alteration in the intrinsic activity of cuprous oxide upon halogen doping. Furthermore, the capacitance measurements of the catalysts revealed a hierarchical sequence: Cu_2_O (3.05 mF cm^−2^), Cu_2_O/Cl (3.44 mF cm^−2^), Cu_2_O/I (4.18 mF cm^−2^), and Cu_2_O/Br (5.17 mF cm^−2^). This sequence indicates a progressive increase in the ECSA and suggests that Cu_2_O/Br manifests a comparatively larger ECSA, thereby exposing an augmented number of active sites for catalysis. Consequently, the catalytic performance of these samples demonstrated a congruent trend with the density of catalytically active sites. This correlation underscores the significant role of ECSA in enhancing the electrocatalytic efficiency of Cu_2_O-based catalysts for ammonia synthesis, thereby illuminating new avenues for optimizing catalytic materials in this domain.

In this advanced analysis, employing X-ray photoelectron spectroscopy (XPS), the study reveals that halogen doping influences the Cu main peak positioning, thereby altering Cu_2_O’s electronic state, as depicted in Figure 7. The electronic states of the copper (Cu) element in cuprous oxide (Cu_2_O) were intricately linked to the ammonia synthesis properties, as depicted in Figure 8a and Appendix A. The impact hierarchy on the electronic state of Cu_2_O is as follows: Cu_2_O < Cu_2_O/Br < Cu_2_O/I < Cu_2_O/Cl. Intriguingly, a correlation was observed between the electronic state variations in Cu_2_O and the ammonia synthesis performance. It was noted that the ammonia synthesis efficacy first increased and then decreased with the varying electronic states of Cu_2_O. This observation aligns with the ‘volcano curve’ principle in reaction chemistry [37,38]. It suggests that Cu_2_O/Br demonstrates peak activity when the adsorption strength of reactive intermediates on Cu_2_O’s surface is neither too strong (as in Cu_2_O/Cl) nor too weak (as in Cu_2_O), hence optimizing ammonia reaction activity.

Another focus of this study is the effect of oxygen vacancy content in halogen-doped Cu_2_O on the ammonia production rate, as shown in Figure 8b and Appendix A. X-ray photoelectron spectroscopy (XPS) analysis reveals that halogen doping significantly impacts the concentration of oxygen vacancies in cuprous oxide. The hierarchy in oxygen vacancy content is observed as Cu_2_O/Cl < Cu_2_O < Cu_2_O/I < Cu_2_O/Br. Intriguingly, the efficiency of ammonia synthesis showed a positive correlation with the increase in oxygen vacancy content. Notably, Cu_2_O/Br emerged as the most effective, underscoring a direct relationship between the augmented oxygen vacancies and enhanced ammonia synthesis performance. This phenomenon highlights the critical role of halogen doping in modulating oxygen vacancy concentration, thereby influencing the overall reactivity and efficiency in the synthesis of ammonia.

## 3. Materials and Method

### 3.1. Reagents

Ascorbic acid (C_6_H_8_O_6_), salicylic acid (C_7_H_6_O_3_), and nitrosoferricyanide (C_5_H_4_FeN_6_Na_2_O_3_∙2H_2_O) were obtained from Aladdin Reagent (Shanghai, China) Co.; sodium hydroxide (NaOH) was obtained from Tianjin Damao Chemical Reagent Factory (Tianjin, China); ethanol (C_2_H_6_O) and potassium bicarbonate (KHCO_3_) were obtained from McLean Reagents (Shanghai, China) Technology Development Co.; sodium citrate (C_6_H_5_Na_3_O_7_∙2H_2_O) was obtained from Tianjin Ding Sheng Xin Chemical Co. (Tianjin, China).

### 3.2. Synthesis of Cuprous Oxide (Cu_2_O) and Cu_2_O/X (X = Cl, Br, I)

This synthesis protocol prepared an initial solution by dissolving 2.0 g of copper sulfate into 55 mL of ascorbic acid solution. This mixture was subjected to continuous agitation for 10 min to ensure forming a uniform solution and designated as Solution 1. After that, 0.5 g of polyvinylpyrrolidone was introduced into Solution 1. The resultant homogenous mixture was then denoted as Solution 2. After this preparation, Solution 2 underwent a 20 min ultrasonication process and was then transferred to a Teflon-lined stainless steel autoclave. This reactor facilitated a reaction process at 120 °C for 4 h. The solution was cooled to ambient room temperature after the reaction. The resultant residue was subjected to a rigorous purification process involving triple washing with absolute ethanol and deionized water to eliminate impurities. The final stage of the procedure involved drying the Cu_2_O microparticles for 5 h at 60 °C in a drying oven, culminating in the acquisition of pure Cu_2_O microparticles, carefully characterized and documented for further analytical studies. 

Utilizing the methodology delineated previously, Solution 2 was synthesized. Subsequently, distinct halogenated variants were prepared by adding 0.05 g of NH_4_Cl, 0.0962 g of NaBr, and 0.1406 g of NaI, respectively, resulting in the formulations designated as Solutions D1, D2, and D3. These solutions underwent a 20 min ultrasonic treatment and were then transferred into a stainless steel reactor, with the interior lined with polytetrafluoroethylene (PTFE). The reactions were conducted at 120 °C for 4 h. Upon completion, the solutions were allowed to cool to ambient room temperature. The precipitated products from each reaction were meticulously collected and subjected to a rigorous purification process involving thrice-repeated washes with anhydrous ethanol and deionized water, ensuring the removal of impurities and unreacted substrates. After the washing steps, the halogenated Cu_2_O microparticles were isolated by drying them in a desiccator at 60 °C for 8 h. This meticulous drying process facilitated the removal of residual moisture, yielding high-purity samples. The final products, characterized by their halogen substituents, were classified as Cu_2_O/Cl, Cu_2_O/Br, and Cu_2_O/I.

### 3.3. Materials Characterization

Physical phase analysis of the catalysts was carried out using a Rigaku mini Flex II X-ray diffractometer of Rigaku Japan, in which the anode target was a copper target, and the X-ray source was Cu Kα rays at a wavelength of *λ* = 0.15406 nm. Under the test conditions, the operating voltage was set at 40 kV and the operating current at 100 mA. The scanning rate of the instrument was set to 8 degrees per minute, and the scanning step was set to 0.02 degrees, while the scanning range was from 20–80 degrees. The morphology and elemental distribution of the catalysts were analyzed by a JEOLJSM-7800F field emission scanning electron microscope equipped with an energy-dispersive X-ray spectrometer (EDS). The ultraviolet-visible (UV-Vis) absorption of the samples is measured by the Shimadzu UV-2600 UV-Vis spectrophotometer. It is based on the absorption characteristics of a substance for light at specific wavelengths and allows the determination of a substance’s concentration, composition, and structural information. The XPS (X-ray photoelectron spectroscopy) characterization of the samples was performed using a PHI5000 Versa Probe instrument, a surface analysis technique used to study the surface chemical state and composition of materials. In XPS characterization, the excitation source is X-rays produced by bombarding an aluminum (Al) anode target with an electron beam of a particular energy. These X-rays are monochromatized by a quartz crystal, leaving behind Al K*α*1 rays with an energy of *hv* = 1486.6 eV. 

### 3.4. Electrochemical Measurements

Electrochemical assessments were meticulously conducted employing a sophisticated CHI880D electrochemical workstation. The apparatus incorporated an H-cell, architecturally bifurcated into a dual-compartment configuration by a selectively permeable Nafion 211 membrane. The preparation of this membrane entailed a series of thermal treatments: initially, it was boiled in ultrapure water for one hour, then incubated in a 5% hydrogen peroxide aqueous solution at 80 °C for an additional hour. This step was succeeded by a one-hour submersion in 0.05 M sulfuric acid and concluded with a three-hour rinse in ultrapure water [39]. The electrode fabrication protocol commenced with the dispersion of 2 mg of the Cu_2_O catalyst (variants: Cu_2_O/Cl, Cu_2_O/Br, Cu_2_O/I) in 475 µL of isopropanol/water in a 2:1 volume ratio solution. This suspension was sonicated for 20 min to achieve a uniform distribution. Subsequently, 25 μL of Nafion solution was integrated and sonicated for an additional 5 min, culminating in the formation of a homogenous ink. A volume of 25 μL of this Cu_2_O catalyst ink was meticulously applied onto a 0.5 cm × 1 cm carbon paper, which underwent a 10 min exposure to light and was preserved in a plastic vial, appropriately labeled with the catalyst identity and test voltage. The catalyst-loaded electrode was secured onto a glassy carbon electrode, serving as the working electrode with a precise catalyst loading of 0.2 mg cm^−2^. The reference electrode was an Ag/AgCl electrode immersed in a saturated potassium chloride solution, complemented by a carbon rod counter electrode. All recorded potentials were normalized to the Reversible Hydrogen Electrode (RHE) scale, employing the conversion formula E_RHE_ = E_Ag/AgCl_ + 0.0591 pH + 0.197. The selected electrolyte for this investigation was a meticulously prepared NO_3_^−^RR solution comprising 10 g of potassium bicarbonate and 10 g of potassium nitrate, precisely calibrated in a 1000 mL volumetric flask. Each anodic and cathodic cell compartment was filled with 30 mL of this electrolyte. During the electrocatalytic process, a constant flow rate of 30 mL min^−1^ was maintained, accompanied by continuous stirring at 400 rpm. This was followed by Linear Sweep Voltammetry (LSV) at a scan rate of 50 mV s^−1^. Subsequently, timed amperometric analyses were performed at different potentials, and current density and charge accumulation were meticulously recorded.

### 3.5. Product Characterization

Standard calibration curves were established to quantify ammonia concentrations utilizing a systematic methodology. A range of solutions with ammonia concentrations varying from 0.2 ppm to 6 ppm were meticulously prepared. A 5 mL vial was employed for each assay, introducing 2.0 mL of the standard ammonia solution. This was followed by the sequential addition of 2.0 mL of Solution A (comprising a 1 M sodium hydroxide solution containing 5% salicylic acid and 5% sodium citrate), 1.0 mL of Solution B (0.05 M sodium hypochlorite), and 0.2 mL of Solution C (1% sodium nitro ferrocyanide). Following a two-hour incubation period in a dark environment at ambient room temperature, the optical properties of the solution were quantitatively assessed at 662nm using UV-vis. Experimental water was employed as the reference standard in this analytical procedure. A standard calibration curve was constructed based on the absorbance data obtained from these measurements, adhering to the experimental results. Furthermore, the urea concentration in question was ascertained using the indanol blue method, which is analogous to the abovementioned steps.

## 4. Conclusions

This research investigated the origin of the electronic-dependent activity for the model Cu_2_O catalyst toward ammonia electrosynthesis with nitrate. XRD, XPS, and SEM characterization of the composites Cu_2_O and Cu_2_O/X (X = Cl, Br, I) confirmed that the modulation of the electronic state and oxygen vacancy content of Cu_2_O was realized by doping with halogen elements (Cl, Br, I). With the increase of electronic state defects in Cu_2_O, the ammonia synthesis performance increased first and then decreased. The Cu_2_O/Br with electronic defects in the middle showed the highest ammonia yield of 11.4 g h^−1^ g^−1^ at −1.0 V (vs. RHE), indicating that the pattern of change in optimal ammonia activity is consistent with the phenomenon of volcano curves in reaction chemistry. In summary, doping affects electronic states and enhances reactivity. This work highlights a promising route to design catalysts for selective NO_3_^−^RR to NH_3_ conversion.

## Figures and Tables

**Figure 1 molecules-29-02261-f001:**
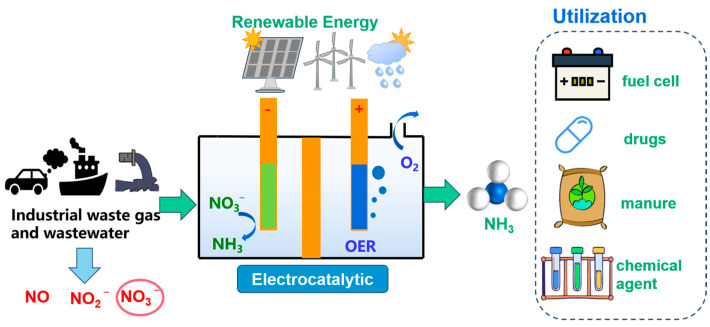
Electrocatalytic ammonia synthesis process.

**Figure 2 molecules-29-02261-f002:**
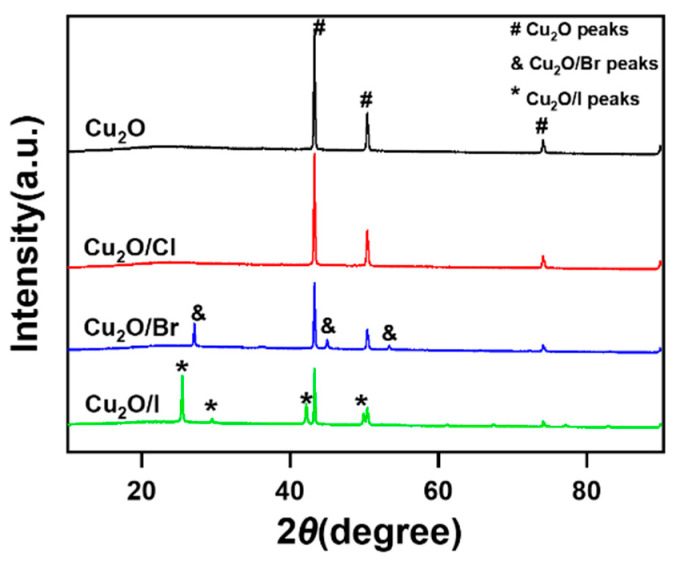
XRD patterns of Cu_2_O, Cu_2_O/X (X = Cl, Br, I).

**Figure 3 molecules-29-02261-f003:**
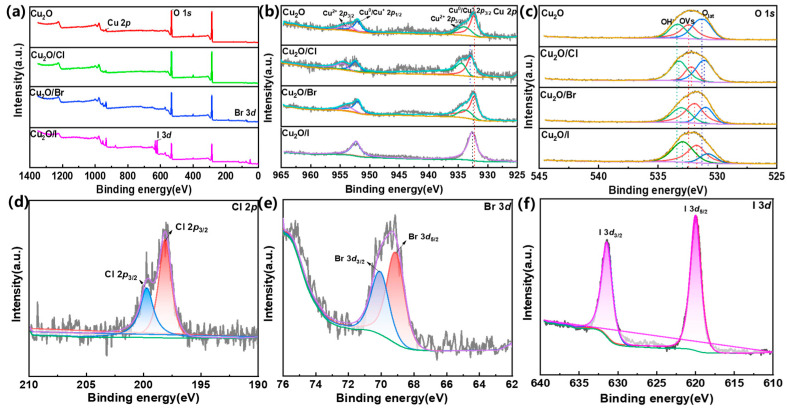
(**a**) General XPS spectra of Cu_2_O and Cu_2_O/X (X = Cl, Br, I); (**b**) XPS plots of Cu peaks of Cu_2_O and Cu_2_O/X (X = Cl, Br, I); (**c**) XPS plots of O peaks of Cu_2_O and Cu_2_O/X (X = Cl, Br, I); (**d**) XPS plots of Cl peak of Cu_2_O/Cl; (**e**) XPS plot of Br peak of Cu_2_O/Br; (**f**) XPS plot of I peak of Cu_2_O/I.

**Figure 4 molecules-29-02261-f004:**
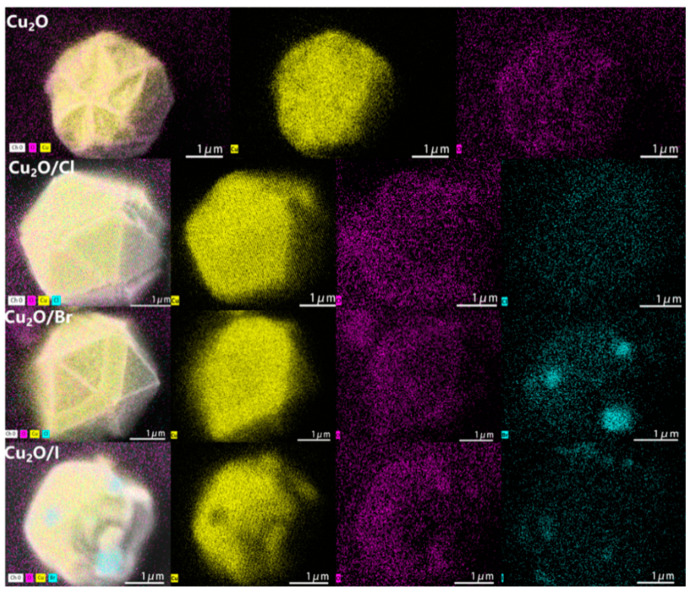
SEM mapping image of Cu_2_O and Cu_2_O/X (X = Cl, Br, I).

**Figure 5 molecules-29-02261-f005:**
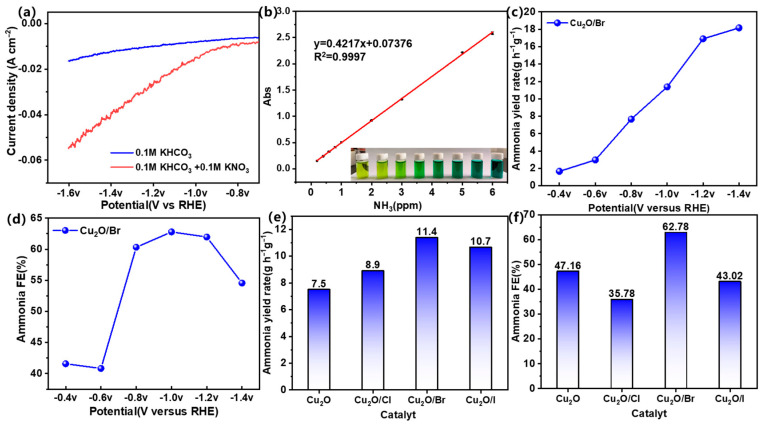
(**a**) LSV plots for electrolyte 0.1 M KHCO_3_ and electrolyte 0.1 M KHCO_3_ + 0.1 M KNO_3_; (**b**) standard curve of ammonia; (**c**) ammonia yield distribution of Cu_2_O/Br catalyst at different voltages; (**d**) Faraday yield distribution of Cu_2_O/Br catalyst at different voltages; (**e**) ammonia yield distribution of Cu_2_O and Cu_2_O/X (X = Cl, Br, I) catalysts at −1.0 V (vs RHE); (**f**) ammonia yield distribution of Cu_2_O and Cu_2_O/X (X = Cl, Br, I) catalysts at −1.0 V (vs RHE) with Faraday efficiency distributions.

**Figure 6 molecules-29-02261-f006:**
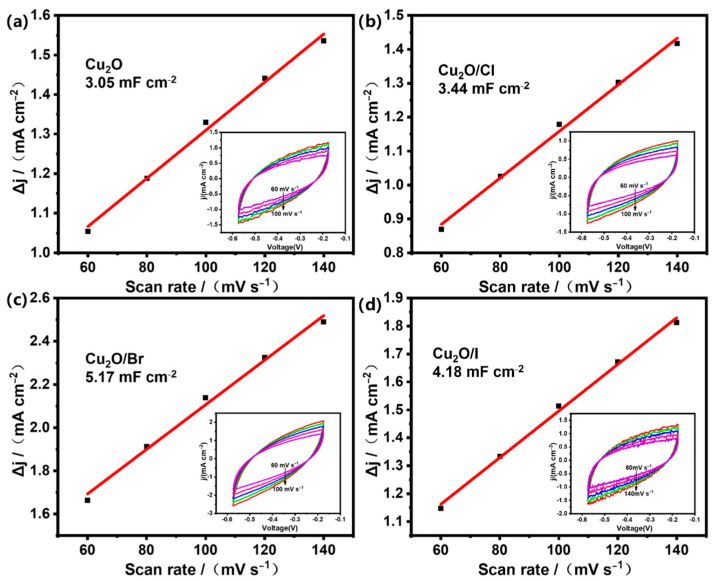
Capacitance current curves and CV curves for Cu_2_O and Cu_2_O/X (X = Cl, Br, I) catalyst samples (**a**–**d**).

**Figure 7 molecules-29-02261-f007:**
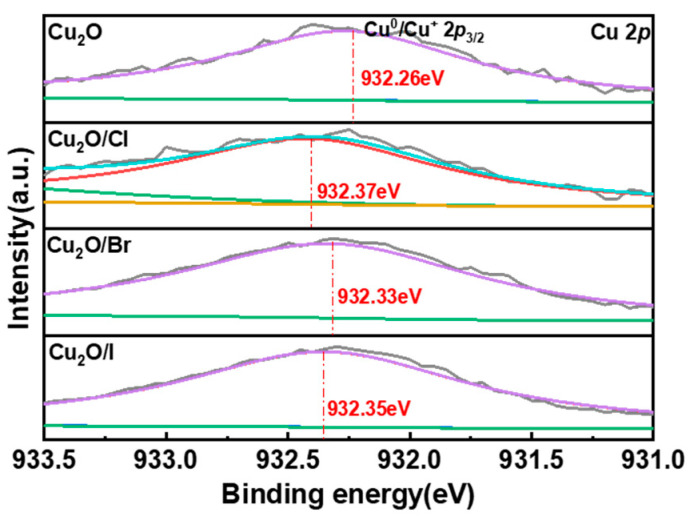
Localized XPS plots of the Cu main peaks of Cu_2_O and Cu_2_O/X (X = Cl, Br, I).

**Figure 8 molecules-29-02261-f008:**
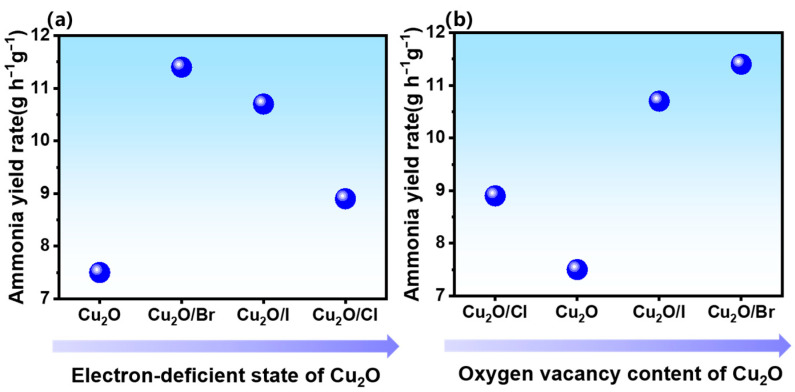
Plot of Cu_2_O electron-deficient state and oxygen vacancy content versus ammonia yield rate.

## Data Availability

The data are available from the corresponding authors upon reasonable request.

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
