# Peer review of "Insights into the Origin of Activity Enhancement via Tuning Electronic Structure of Cu2O towards Electrocatalytic Ammonia Synthesis"

_molecules, 2024, doi:10.3390/molecules29102261_

Round 1

Reviewer 1 Report

Comments and Suggestions for Authors

The manuscript presents an intriguing exploration of electronic structure modulation in Cu2O catalysts for electrocatalytic ammonia synthesis, focusing on the doping of Cl, Br and I. Introducing a novel approach, the study highlights the correlation between electronic state defects in Cu2O and ammonia synthesis activity. Utilizing characterization techniques such as XRD, XPS, and SEM, alongside electrochemical studies, strengthens the significance of the findings for Cu2O-halogen catalysts in ammonia synthesis. However, a few points need to be clarified:

1. Provide high-resolution SEM images of the synthesized Cu2O-halogens to know the structural changes with change of halogens. 2. What is the concentration of each halogen in the Cu2O-halogen catalysts?  3. How long the ammonia production was carried out for each catalyst mentioned in the figure 5d? 4. A thorough comparison with existing methodologies or materials for electrocatalytic ammonia synthesis would be beneficial.  5. Additionally, discussing the stability/long-term performance of the synthesized Cu2O-halogen catalysts for ammonia synthesis would further enhance the study's impact. 

Addressing these suggestions would enrich the significance of the research, ultimately advancing the field of electrocatalytic ammonia synthesis.

Author Response

For research article

Response to Reviewer 1 Comments

1. Summary

Thank you for your letter and the reviewers’ comments on our manuscript (molecules-2984246). These comments are very valuable and helpful for us to revise the manuscript and improve our work, as well as important guidance for our further research.

Firstly, we have carefully revised the manuscript, including the use of English, and added some new data in the revised version. All revision traces are marked in Red. Please refer to the revised manuscript for all details.

Secondly, point-to-point responses to the comments and suggestions are given below.

2. Point-by-point response to Comments and Suggestions for Authors

The manuscript presents an intriguing exploration of electronic structure modulation in Cu2O catalysts for electrocatalytic ammonia synthesis, focusing on the doping of Cl, Br, and I. Introducing a novel approach, the study highlights the correlation between electronic state defects in Cu2O and ammonia synthesis activity. Utilizing characterization techniques such as XRD, XPS, and SEM, alongside electrochemical studies, strengthens the significance of the findings for Cu2O-halogen catalysts in ammonia synthesis. However, a few points need to be clarified:

Comments 1: Provide high-resolution SEM images of the synthesized Cu2O-halogens to know the structural changes with changes of halogens.

Response 1: Thanks for your advice. As suggested, the high-resolution SEM and TEM images of the synthesized Cu2O-halogens have also been added in Figure S1. Clearly, both Cu2O and Cu2O/X (X = Cl, Br, I) exhibit irregular shapes, and the doping of halogens has little effect on the Cu2O morphology.

Figure S1 (a) and (b) represent SEM images of Cu2O and Cu2O/X (X = Cl, Br, I) at 50 μm and 1 μm; (c) TEM images of Cu2O and Cu2O/X (X = Cl, Br, I)

Comments 2: What is the concentration of each halogen in the Cu2O-halogen catalysts?

Response 2: Thank you very much for your carefulness. The following is what we derived the concentration of each halogen in the Cu2O-halogen catalysts from XPS full spectrum analysis.

Table S1. The concentration of each halogen in the Cu2O-halogen catalysts

Catalyst

halogen

Atomic (%)

Cu2O/Br

Br

3.27

Cu2O/Cl

Cl

1.83

Cu2O/I

I

3.44

Comments 3: How long the ammonia production was carried out for each catalyst mentioned in the figure 5d?

Response 3: We thank the reviewer for your carefulness. During the electrocatalytic process, the ammonia production time for each catalyst is 30 min, which has been revised in the manuscript.

Comments 4: A thorough comparison with existing methodologies or materials for electrocatalytic ammonia synthesis would be beneficial.

Response 4: Thanks for your valuable suggestions. As suggested, a thorough comparison with existing methodologies or materials for electrocatalytic ammonia synthesis was made in the revised manuscript. Clearly, the Cu2O/Br has excellent ammonia synthesis rates (exceeding most reported catalysts) and appropriate Faraday efficiencies (Figure.S2, Table S2).

The following test has been added to the revised manuscript to address the reviewer’s comments:

Moreover, a thorough comparison with existing methodologies or materials for electrocatalytic ammonia synthesis was made in this paper. Clearly, the Cu2O/Br has excellent ammonia synthesis rates (exceeding most reported catalysts) and appropriate Faraday efficiencies (Figure.S2, Table S2).

Figure S2 Comparison of electrocatalytic NO3-RR performance for Cu2O/Br with other electrocatalysts under ambient conditions

Table S2 Comparison of electrocatalytic NO3-RR performance for Cu2O/Br with other electrocatalysts under ambient conditions

Catalyst

Electrolyte

Faradaic efficiency (%)

NH3 Yield rate (g h-1 g-1)

Refs

Cu2O/Br

0.1M KHCO3+

0.1M KNO3

62.78

11.4

This work

Fe-N/P-C

0.1M KOH+

0.1M KNO3

90.3

17.98

1

Fe single atom

0.1M K2SO4+

0.5M KNO3

75

20

2

Cu-Pd/C

0.1M KOH+

0.01M KNO3

62.3

0.22

3

RuxOy

0.1M Na2SO4+

200mg/L NO3-

73

0.274

4

In-S-G

1M KOH+

0.1M KNO3

75

3.74

5

M-NDs/Zr-MOF

0.1M Na2SO4+

500ppm NO3-

58.1

4.88

6

Cu/Cu2+1O

0.5M K2SO4+

50mg/L NO3-

87.07

0.576

7

Cu-NPs

0.5M K2SO4+

50ppm NO3-

81.99

0.781

8

Fe/Cu-HNG

1M KOH+

0.1M KNO3

92.51

0.018

9

Ar-40/Cu2O

0.5M Na2SO4+

200ppm NO3-

85.78

1.188

10

Cu@CuHHTP

0.5M Na2SO4+

500ppm NO3-

67.55

3.68

11

Cu nanosheets

0.1M KOH+

0.01M KNO3

99.7

0.39

12

Co3O4@NiO

0.5M Na2SO4+

200ppm NO3-

54.97

0.118

13

Comments 5: Additionally, discussing the stability/long-term performance of the synthesized Cu2O-halogen catalysts for ammonia synthesis would further enhance the study's impact.

Response 5: Thank you for your useful and necessary comments. As you suggested, we have supplemented the stability electrocatalysis experiment below, confirming the excellent electrochemical stability of Cu2O/Br.

The following test has been added to the revised manuscript to address the reviewer’s comments:

…The stability electrocatalysis experiments of the prepared catalysts were carried out and confirmed the excellent electrochemical stability of Cu2O/Br (Figure. S3).

Figure S3. Stability test of ammonia electrosynthesis on Cu2O/Br at -1.0 V over six continuous cycles (30 min/cycle).

Reviewer 2 Report

Comments and Suggestions for Authors

I want to thank the authors for their excellent work. The spectroscopic analysis in this manuscript is commendable, and I am pleased to accept this version after major revisions. Please find the following points

1. References: The literature contains some iron-based molecular catalysts that can be included in the introduction.

2. Figure 2 captions should be rewritten as XRD patterns, not XRD plots.

3. I would like to thank the authors for Figure 8. It's a nice visual representation of what is going on. However, having more insight into my concerns about the following points would be more appropriate.

Given the author's discovery that oxygen vacancies are the main determining factor of the high reactivity of bromo analogs, I am intrigued. Would it be possible to use XPS to compare the intensity of halogen and oxygen in this series of materials? It can provide an idea of the amount of halogen present in the material, which is tuning the electronic states of the species. This could potentially provide further confirmation of your argument.

 Although the Authors reasonably explain the presence of Cu2+ in the XPS data. I am sure those things can easily be avoided by maintaining inert conditions. I will suggest repeating this experiment at least for their bromo material, and if there is an improvement in the reactivity or not since Cu1+ is doing chemistry here, not cu2+

4. I doubt the linear fitting of the data in Figure 6, especially for their two more reactive bromo and iodo species. They seem exponential in nature. The author should try increasing the scan rate to at least 150- 200 mv/s to check its linearity.

5. Finally, I fully agree with the author's conclusion that doping affects electronic states and enhances reactivity. However, the authors should try to add the above point to the discussion to strengthen their conclusion. 

Author Response

For research article

Response to Reviewer 2 Comments

1. Summary

Thank you for your letter and the reviewers’ comments on our manuscript (molecules-2984246). These comments are very valuable and helpful for us to revise the manuscript and improve our work, as well as important guidance for our further research.

Firstly, we have carefully revised the manuscript, including the use of English, and added some new data in the revised version. All revision traces are marked in Red. Please refer to the revised manuscript for all details.

Secondly, point-to-point responses to the comments and suggestions are given below.

2. Point-by-point response to Comments and Suggestions for Authors

I want to thank the authors for their excellent work. The spectroscopic analysis in this manuscript is commendable, and I am pleased to accept this version after major revisions. Please find the following points.

Response: We thank the reviewer for the positive comments. We have addressed these questions to improve the quality of this work and the details are shown below.

Comments 1: References: The literature contains some iron-based molecular catalysts that can be included in the introduction.

Response 1: Thanks a lot for the reviewer’s valuable consideration. We have supplemented the iron-based molecular catalysts mentioned in Refs and described them in detail in the Introduction.

Wang et al. reported that Fe single atom catalyst effectively prevents the N-N coupling step required for N2 due to the lack of neighboring metal sites, promoting ammonia product selectivity. The maximal ammonia Faradaic efficiency of ~ 75% and a yield rate of up to ~ 20,000 μg  h−1 mgcat−1[5]…

Comments 2: Figure 2 captions should be rewritten as XRD patterns, not XRD plots.

Response 2: Thanks for pointing out the mistake. We have modified the title of Figure. 2 from XRD plots to XRD patterns as shown below.

Figure 2. XRD patterns of Cu2O, Cu2O/X (X = Cl, Br, I)

Comments 3: I would like to thank the authors for Figure 8. It's a nice visual representation of what is going on. However, having more insight into my concerns about the following points would be more appropriate. Given the author's discovery that oxygen vacancies are the main determining factor of the high reactivity of bromo analogs, I am intrigued. Would it be possible to use XPS to compare the intensity of halogen and oxygen in this series of materials? It can provide an idea of the amount of halogen present in the material, which is tuning the electronic states of the species. This could potentially provide further confirmation of your argument.

Response 3: Thanks for your careful and valuable comment. We analyzed the relationship between the copper main peak offset and ammonia yield for the four catalysts by XPS. It was noted that the ammonia synthesis efficacy first increased and then decreased with the offset Cu main peak of catalysts. This observation aligns with the 'volcano curve' principle in reaction chemistry. It suggests that Cu2O/Br demonstrates peak activity when the adsorption strength of reactive intermediates on Cu2O's surface is neither too strong (as in Cu2O/Cl) nor too weak (as in Cu2O), hence optimizing ammonia reaction activity, as shown in S4 (a). We calculated the link between oxygen vacancy content and ammonia yield by XPS-O1s. The hierarchy in oxygen vacancy content is observed as Cu2O/Cl < Cu2O/I < Cu2O/Br. Intriguingly, the efficiency of ammonia synthesis showed a positive correlation with the increase in oxygen vacancy content, as shown in S4 (b).

Figure S4. Plot of catalysts Cu main peak offset and oxygen vacancy content versus ammonia yield rate.

Although the Authors reasonably explain the presence of Cu2+ in the XPS data. I am sure those things can easily be avoided by maintaining inert conditions. I will suggest repeating this experiment at least for their bromo material, and if there is an improvement in the reactivity or not since Cu1+ is doing chemistry here, not Cu2+.

Response 3: Thank you very much for your valuable comments. We apologize for not being able to solve this problem perfectly. During the electrochemical performance test, we do our best to keep the catalyst in N2 inert condition, but inevitably the catalyst is exposed to air during the catalyst synthesis and XPS delivery and testing.

Comments 4: I doubt the linear fitting of the data in Figure 6, especially for their two more reactive bromo and iodo species. They seem exponential in nature. The author should try increasing the scan rate to at least 150- 200 mv/s to check its linearity.

Response 4: We thank the reviewers for pointing out the errors. We have modified the capacitance current curves and CV curves for Cu2O and Cu2O/X (X = Cl, Br, I) by increasing the scan rate to 60-140 mv/s. We have also modified the text as follows. The following test has been added to the revised manuscript to address the reviewer’s comments:

Cu2O (3.05 mF cm-²), Cu2O/Cl (3.44 mF cm-²), Cu2O/I (4.18 mF cm-²), and Cu2O/Br (5.17 mF cm-²)…

Figure 6. Capacitance current curves and CV curves for Cu2O and Cu2O/X (X = Cl, Br, I) catalyst samples

Comments 5: Finally, I fully agree with the author's conclusion that doping affects electronic states and enhances reactivity. However, the authors should try to add the above point to the discussion to strengthen their conclusion.

Response 5: Thanks for your careful comment. Here is the “conclusion” as I have modified it after taking your advice.

This research investigated the origin of the electronic-dependent activity for model Cu2O catalyst toward ammonia electrosynthesis with nitrate. XRD, XPS, and SEM characterization of the composites Cu2O and Cu2O/X (X = Cl, Br, I) confirmed that the modulation of the electronic state and oxygen vacancy content of Cu2O was realized by doping with halogen elements (Cl, Br, I). With the increase of electronic state defects in Cu2O, the ammonia synthesis performance increased first and then decreased. The Cu2O/Br with middle electronic defect showed the highest ammonia yield of 11.4 g h-1 g-1 at -1.0 V (vs. RHE), indicating that the pattern of change in optimal ammonia activity is consistent with the phenomenon of volcano curves in reaction chemistry. In summary, doping affects electronic states and enhances reactivity. This work highlights a promising route to design catalysts for selective NO3-RR to NH3 conversion.

Round 2

Reviewer 1 Report

Comments and Suggestions for Authors

Authors have addressed all my concerns and comments and the manuscript can be published.

Reviewer 2 Report

Comments and Suggestions for Authors

I am satisfied with all the answers from the authors in this revised version. I would like to accept this manuscript in this present version.